# Piezoelectric Energy Harvester Response Statistics

**DOI:** 10.3390/mi14020271

**Published:** 2023-01-20

**Authors:** Oleg Gaidai, Yu Cao, Yihan Xing, Junlei Wang

**Affiliations:** 1Shanghai Engineering Research Center of Hadal Science and Technology, College of Engineering Science and Technology, Shanghai Ocean University, Shanghai 201306, China; 2Department of Mechanical and Structural Engineering and Material Sciences, University of Stavanger, N-4036 Stavanger, Norway; 3School of Mechanical and Power Engineering, Zhengzhou University, Zhengzhou 450001, China

**Keywords:** experiment, piezoelectric energy harvesting, extreme values, galloping, bivariate statistics

## Abstract

Safety and reliability are essential engineering concerns for energy-harvesting installations. In the case of the piezoelectric galloping energy harvester, there is a risk that excessive wake galloping may lead to instability, overload, and thus damage. With this in mind, this paper studies bivariate statistics of the extreme, experimental galloping energy harvester dynamic response under realistic environmental conditions. The bivariate statistics were extracted from experimental wind tunnel results, specifically for the voltage-force data set. Authors advocate a novel general-purpose reliability approach that may be applied to a wide range of dynamic systems, including micro-machines. Both experimental and numerically simulated dynamic responses can be used as input for the suggested structural reliability analysis. The statistical analysis proposed in this study may be used at the design stage, supplying proper characteristic values and safeguarding the dynamic system from overload, thus extending the machine’s lifetime. This work introduces a novel bivariate technique for reliability analysis instead of the more general univariate design approaches.

## 1. Introduction

It is common to extract energy from piezoelectric materials. Piezoelectric materials have been investigated for decades. Several recent developments have been in harvesting electrical energy with these materials [1,2]. Energy harvesters are essential to modern offshore green-energy engineering [3]. Therefore, a proper experimental study and safety and reliability analysis are of practical engineering importance. Researchers in energy harvesting and sensing are becoming increasingly interested in wind energy since it is a renewable and widely dispersed energy. This article considers a device that is 200 × 25 × 0.5 mm^3^. However, micro/nano-wind energy harvesters are also of engineering interest and advocate a reliability approach that is of a general purpose and applicable to any scale. The micro-scale wind energy is harvested using piezoelectric, electrostatic, and triboelectric energy harvesters.

In recent years, low-frequency energy-harvesting technology from the ambient environment has shown its advantages. Such technologies were developed to provide a power supply for some low-power cost devices, e.g., Wireless sensor networks (WNS) and MEMS (micro-electromechanical systems) [4]. Piezoelectric vibration energy Harvesting (PVEH) has shown its advantage in transferring ambient mechanical vibratory energy into electrical energy [5,6]. Harvesting energy from flow-induced vibrations (FIVs) has shown its advantage as its aero-instability occurs in the form of resonance vibration, which is a benefit for designing prototypes. FIVs among various vibrations include galloping, vortex-induced vibrations (VIV), wake-induced vibration (WIV), etc.

A series of studies on experimental, numerical, and theoretical methods have been conducted in the recent decade. Study 1 introduced an aero-instability coupling model, mechanical vibration, and electrical circuit. Ref. [2] presented a CFD study to study the VIV piezoelectric energy harvester (VIVPEH). Ref. [3] studied theoretical aspects of the VIV piezoelectric energy harvester using Galerkin’s algorithm and then conducted a series of linear and non-linear experiments to study the optimisation method and multifunctional VIVPEH. In a recent study [4], authors reported a successful transition from VIVPEH to GPEH (galloping-based piezoelectric energy harvester) by adding two Y-shaped attachments. For recent applications of aero-flow piezoelectric energy-harvesting systems discussing galloping and the VIV phenomenon [7,8].

Most previous works focused on energy harvester (EH) performance enhancements. Still, few of them paid attention to practical design issues such as the EH’s extreme response and selecting proper design values. Studying the extreme performance of the EH and avoiding adverse working conditions are essential parts of the modern design process. For example, to predict the fatigue and damage of the piezoelectric energy harvester, authors in [9,10,11] investigated the reliability and durability of a P1 MFC transducer subjected to base excitations. In [10], authors reported experimental results and found that an acceleration level of 0.4–0.5 g was harmful to the energy harvester. In [12], authors have studied the performance of an EH with DuraAct P-876.A12 piezo-sheet on the railway, and it was reported that the EH failed after 106 cycles under the acceleration of 1 g. Next, [13] investigated EH long-term fatigue behaviour with a P-2 MFC sheet; both Finite Element Analysis (FEA) and laboratory experiments were conducted. In [14], a considerable drop in the EH output values occurred after two million cycles under base acceleration levels of 0.4, 0.5, and 0.6 g, respectively, causing initial damage in the piezo transducer. [15] were first to propose a fluid flow nozzle-based energy harvester. It was stated that the harvester beams were damaged within minutes under a flow rate of 15 L/min. A steel shim-reinforced harvester beam endured a flow rate of 16.5 L/min for about 40 min. No damage was observed under a flow rate of 9.1 L/min after 9 h of testing. Furthermore, no related studies focused on EH durability for flow-based energy harvesters, e.g., GPEH.

There was nearly no research focusing on experimental testing of the dynamic performance of wind-induced vibration energy harvesting with respect to its reliability. Recent studies usually focused on the behaviour of the energy-harvesting beam undergoing fatigue loading due to specific excitations to the estimate durable EH lifetime. Still, few studies focused on the EH extreme-values analysis and, thus, the safety and reliability of its working conditions.

The practicality of this study may be tied to “characteristic” (or “design”) values that have to be determined during the device design stage, and therefore being an important key-input for a designer.

This paper studies the extreme value statistics of a GPEH system dynamic response as part of broader reliability research. A distinctive feature of this study lies in its bivariate approach, while most extreme-value statistical techniques are univariate. To achieve the latter purpose, experimental and theoretical studies have been conducted to investigate the dynamic performance of a specific galloping energy harvester.

## 2. Methodology and Analysis

Figure 1 presents the experimental setup and the details of the bluff body. A round cross-section wind tunnel was used to test the performance of the GPEH. As shown in Figure 1, the wind tunnel with a diameter of 400 mm can produce uniform incoming flow by installing a honeycomb structure inside the settling chamber. The produced wind speed was within a range of 1 ≤ *U* ≤ 6 m/s; note that 6 m/s wind speed may induce VIV strong enough to cause EH damage. Given specific wind-speed probabilities, the discrete constant wind speeds may then be plugged into this in situ distribution. A piezoelectric sheet (Model: PZT-5, JiaYeShi., China) of 30 × 20 × 0.5 mm^3^ was equipped on a substrate made of pure aluminium with dimensions of 200 × 25 × 0.5 mm^3^ to construct the piezoelectric cantilever [6,7,8]. The clamped capacitance of the piezoelectric transducer $C_{P}\;$ was 30.5 nF. A circular-sectioned bluff body was connected at the free end of a piezoelectric cantilever, as shown in Figure 1. The bluff body was made of hard foam with a length of 0.118 m and a diameter of 0.032 m. The prototyped piezoelectric cantilever damping ratio *ζ* was measured using the logarithmic decrement technique. The wind speed *U* is measured using a hot-wire anemometer (Model: 405i, Testo Co., Sparta, NJ, USA). The electrical module comprises an electrical circuit which converts and rectifies the generated voltage V. The voltage output was measured using a digital oscilloscope (Model: DS1104S, RIGOL., Suzhou, China).

One must point out that the target of a GPEH is designed to harvest low-speed wind energy, practically *U* ≤ 6 m/s. If the wind speeds are high (≥7 m/s), a traditional propeller generator is a more suitable engineering choice.

## 3. Results

This section presents the voltage-force dataset bivariate ACER2D (averaged conditional exceedance rate, two-dimensional)-method statistical results obtained from the experimental record. By force in this study, authors meant the horizontal wind force acting on the GPEH bluff body part. Due to vortex separation, this force oscillates; see Figure 1. As the underlying statistical data set, stationary, two-minute-long time series recordings matching diverse ambient wind conditions were used. With a time step of *dt* = 0.02 s, 10 independent stationary realisations of the voltage-time series were monitored for each environmental state. The total number of different environmental conditions modelled in the experiment was six; six different wind speeds were assigned their probabilities of occurrence according to their potential in situ wind speed distribution. For more details on bivariate ACER2D functions [16,17,18,19,20,21,22,23,24,25,26,27,28,29,30,31,32,33,34,35].

Piezoelectric energy harvesters always aim to scavenge energy from low wind speeds because when the wind speeds reach higher values, the energy harvesting efficiency will dramatically decrease [30,31]. Thus, a stopper can be designed for the prototype to prevent damage under high wind speeds. Due to that, the vibration will be suppressed with the large vibration amplitude caused by high wind speeds by the stopper; see [9]. Note that energy-harvester operating wind speeds can be controlled in the region of 1–6 m/s. The aerodynamic force acting on the bluff body time series was obtained from the measured voltage time series using the following linear equation:(1)$$F=\frac{C_{p}}{\theta }\left(M\ddot{V}+C\dot{V}+KV\right)+\theta V$$
with $C_{p},\;M,\;C,\;K,\;\theta$ being the physical device-specific parameters [16]. In this study, a bivariate random process Z(t)=(X(t),Y(t)) has been considered, consisting of voltage and force processes X(t),Y(t), with the first component (voltage X=V) being measured and the second component (force Y=F) being measured synchronously, over a certain time span (0,T). Let one assume that samples $\left(X_{1},Y_{1}\right),\ldots ,\left(X_{N},Y_{N}\right)$ are taken at *N*-equidistant discrete time moments $t_{1},\ldots ,t_{N}$ within the measurement time span (0,T) [32,33].

This paper studies the bivariate joint cumulative distribution function (CDF) $P\left(\xi ,\eta \right):=\;\mathrm{Prob}\;\left({\hat{X}}_{N}\leq \xi ,{\hat{Y}}_{N}\leq \eta \right)$ of the 2D vector $\left({\hat{X}}_{N},{\hat{Y}}_{N}\right)$, with components ${\hat{X}}_{N}=\max\left\{X_{j}\;;j=1,\ldots ,N\right\}$ and ${\hat{Y}}_{N}=\max\left\{Y_{j}\;;j=1,\ldots ,N\right\}$. In this paper ξ and η are the voltage and force values, respectively, measured synchronously at the same in situ location and the same device.

Figure 2 presents the correlation between measured voltage and the energy harvester’s corresponding force. Generally, based on various experimental measurements, the wind speed-probability density function follows a Weibull distribution [16]
(2)$$H\left(\bar{U}\right)=\frac{\delta }{\beta }{\left(\frac{\bar{U}}{\beta }\right)}^{\delta -1}\cdot e^{-{\left(\frac{\bar{U}}{\beta }\right)}^{\delta }}$$
where β and δ are known as the scale and shape parameters, respectively. $\bar{U}=\frac{U}{\omega_{n}D}$ represents reduced wind speed, with U being the wind speed itself, $\omega_{n}=\sqrt{k/m}$ the natural vibration frequency of the mechanical oscillator, *k* and *m* are mass and stiffness, respectively, and *D* is the GPEH downstream cylinder diameter; see Figure 1. The latter kind of distribution, having shape factors between δ one and three, is typically found in nature.

Figure 3 presents the Weibull probability density functions used to model wind-speed distribution. For this study, the Weibull scale parameter β was chosen to be equal to five, as it supports lower wind speeds since lower wind speeds are inherent in the energy-harvester design used in this experimental study. The distribution choice is typical for characterising the response of galloping energy harvesters using in situ wind statistics [17].

It is seen from Figure 2 and Equation (1) that force is strongly linearly correlated with its underlying voltage due to the smallness of $\frac{C_{p}}{\theta }$; however, in the extreme distribution tail, the non-linear relationship dominated, as is again seen from Figure 2 and Equation (1) due to the increased values of $\dot{V}$, $\ddot{V}$. The latter phenomenon requires an accurate non-linear statistical analysis of the probability-distribution tails. As it will be seen from the current study, the ACER2D method is well suitable for the abovementioned task [30,36].

Figure 4 presents the ACER (averaged conditional exceedance rate) marginal statistics voltage versus corresponding force for energy harvester, with the ACER function conditional parameter k being responsible for data de-clustering and indicates the converged level of ACER*_k_* functions [25,26,27,28,29]. It is seen from Figure 4 (right) that the force acting on the energy harvester exhibits a non-linear change in the distribution tail pattern near force F≈55 mN, therefore the seeming linear correlation visible in Figure 2 does not hold for extreme statistics. The inherent ACER function advantage lies in its ability to accurately approximate exact extreme-value distribution inherent in the data set, both in the univariate and the bivariate case [31,32,33,34,35].

Figure 5 shows contour plots of (A) the optimised AL (Asymmetric logistic) $A_{k}\left(\xi ,\eta \right)$ and (B) optimised GL (Gumbel logistic) core function $G_{k}\left(\xi ,\eta \right)$ that match the empirical bivariate ACER2D function ${\hat{ℰ}}_{k}\left(\xi ,\eta \right)$, k=3 for AL, GL copula definitions [22,23,24]. The level of conditioning k=3 was chosen since ACER2D functions ${\hat{ℰ}}_{k\;}$ exhibited tail convergence to ${\hat{ℰ}}_{3}$. The negative numbers depicted over contours in Figure 5 show decimal logarithmic scale probability levels for CDF P(ξ,η); see Appendix. Figure 5 suggests that the empirical bivariate ACER2D-surface ${\hat{ℰ}}_{3}$ partially matches the correlation between the measured voltage and force, while optimally fitted AL $A_{3}$ and GL $G_{3}$ produced smooth bivariate contours. Note that fully correlated data would yield contour lines consisting of only horizontal and vertical lines.

Given that the laboratory-measured data set is quite limited, Figure 5 reasonably matches the estimated bivariate ACER2D and the optimised Asymmetric logistic (AL) and Gumbel logistic (GL) copulas. Thus, the 2D-optimised copula model, based on asymptotically matched 1D marginals, can be used to parametrically represent the bivariate ACER2D surface corresponding to the measured voltage-force data set.

Figure 6 presents return periods of 2, 5, and 10 years per contour line. Return periods of the order of 10 years are of practical relevance for the energy harvester design [16,17,18], as well as for various offshore energy installations such as offshore windfarm vessels [19], floating wind turbines [25], and wave-height and wind-speed bivariate statistics [32,33,34].

## 4. Discussion

In most practical engineering cases, the 1D univariate design points are used, while the 2D bivariate coupling is often not considered. Thus, it is appropriate to suggest an alternative design-point selection (basically 2D correction, with respect to 1D design point) based on bivariate statistical analysis. Figure 6 suggests a design-point 2D selection. Note that 2D design points are less conservative than 1D (the latter are located at intersections of horizontal and vertical marginal 1D lines), leading to a cheaper but equally safe engineering design.

Note that open-circuit voltage is not the only important EH safety response. The short-circuit current of the harvester is of crucial importance as well. However, this study aims to illustrate bivariate statistical methodology; thus, choosing any alternative couple of responses will be possible.

Underlying statistical data distribution tail non-regularity was seen in Figure 4 in the form of a right-force (or equivalent acceleration) component. Still, the proposed bivariate methodology coped well with the challenge, yielding smooth predicted contours.

The above-described scheme may be well used for a wide variety of energy-harvesting models and various types of environmental conditions.

## 5. Conclusions

This paper studied energy harvester dynamic response during operation in random wind conditions. This study suggests a design-point modification by applying the recently developed bivariate ACER2D method. The latter method was applied to a specific voltage-force dataset obtained from laboratory experiments. The bivariate ACER2D technique is based on a generalisation of the univariate ACER1D technique of an average conditional exceedance rate to a bivariate data set case. This paper studied the joint distribution of two correlated random variables: energy harvester voltage and force. Extreme-value distribution tail quantiles were obtained by utilising bivariate copulas.

Regarding the optimal structural design concept, it is essential to note that performing only univariate design-value analysis may be over-conservative compared to bivariate or multivariate analysis. The latter may be a practical argument for an optimal design routine. The advantages of the presented bivariate technique are:

A broad scope of coupled data can be studied, obtained using the numerical Monte Carlo simulation and experimentally measured.

Even relatively short data records can yield meaningful design results, provided proper statistical methods are applied.

The advantage of the proposed methodology is that a relatively short experimental data record can still yield meaningful statistical and design results, provided proper statistical methods are applied. By “relatively short data”, the authors refer to the ability of the suggested methodology to extrapolate a probability tail of a few orders of magnitude on a decimal logarithmic scale. The limitation of the suggested reliability technique lies within underlying statistical assumptions and the corresponding data set quality. The latter is relevant to any practical statistical method.

As for future studies, the authors intend to introduce a multivariate reliability approach suitable for analysing the safety of the complex energy harvester dynamic system as a whole.

## Figures and Tables

**Figure 1 micromachines-14-00271-f001:**
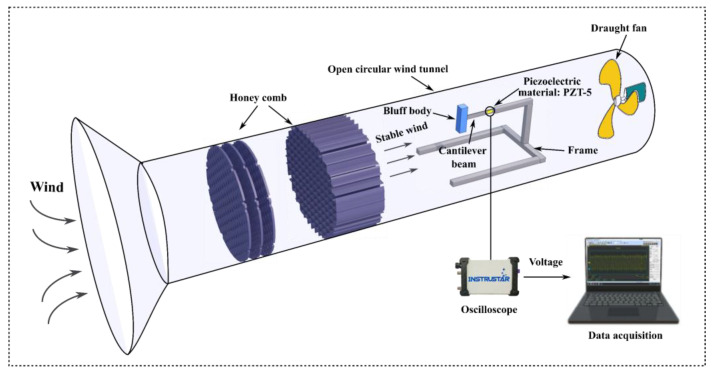
The wind tunnel experiment setup for stable wind [6].

**Figure 2 micromachines-14-00271-f002:**
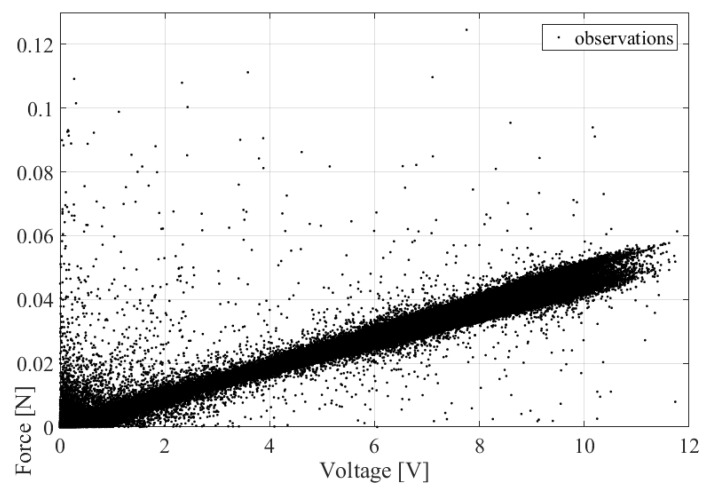
Measured voltage versus corresponding force for energy harvester.

**Figure 3 micromachines-14-00271-f003:**
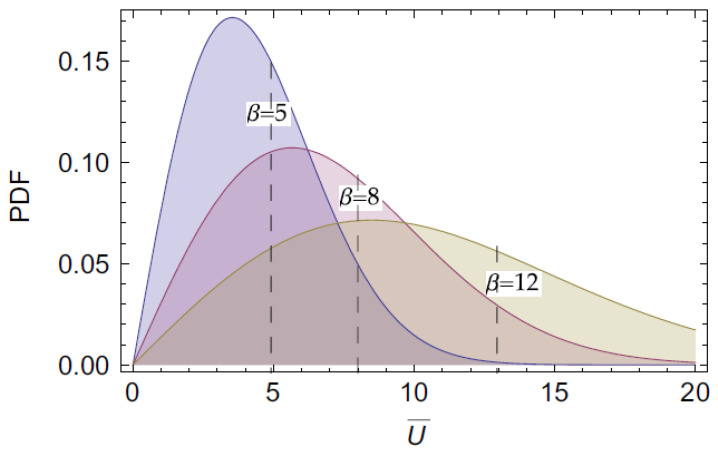
Wind speed in m/sec distribution examples, Weibull probability density functions, from [16].

**Figure 4 micromachines-14-00271-f004:**
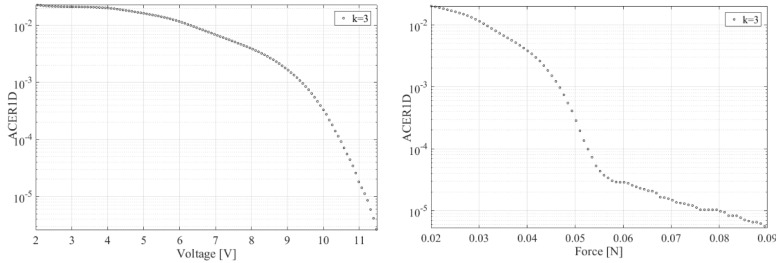
ACER1D marginal statistics voltage versus corresponding force for energy harvester, voltage left, force right. Decimal logarithmic scale on the vertical axis; k is the ACER function conditional parameter.

**Figure 5 micromachines-14-00271-f005:**
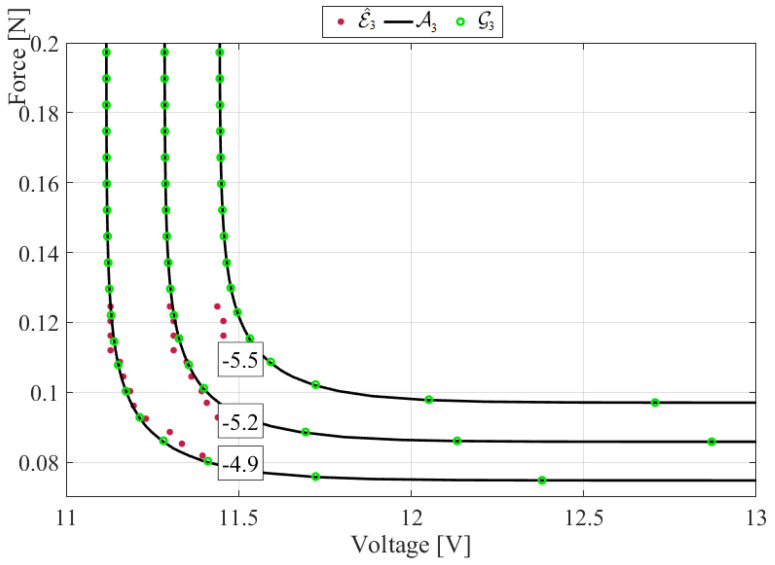
Optimised Asymmetric logistic $A_{3}\left(\xi ,\eta \right)$ (—) contour plot and (A) empirically estimated ${\hat{ℰ}}_{3}\left(\xi ,\eta \right)$ (•), (B) optimised Gumbel logistic $G_{3}\left(\xi ,\eta \right)$ (∘). Negative numbers reflect probability levels on a decimal logarithmic scale.

**Figure 6 micromachines-14-00271-f006:**
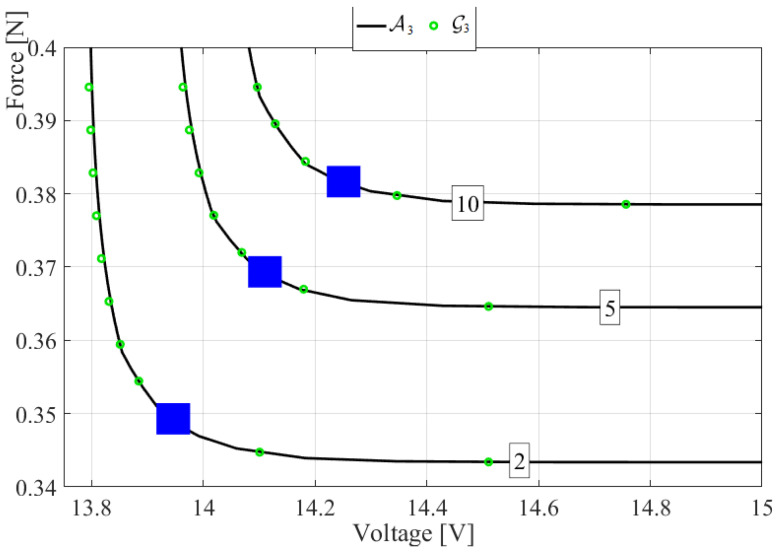
Return periods contour plot for (A) optimised Gumbel logistic $G_{3}\left(\xi ,\eta \right)$ (∘) and (B) optimised Asymmetric logistic $A_{3}\left(\xi ,\eta \right)$ (—) surfaces. Boxes highlight return periods in years. Solid squares indicate suggested 2D design points.

## Data Availability

Not applicable.

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
