# Peer review of "Piezoelectric Energy Harvester Response Statistics"

_micromachines, 2023, doi:10.3390/mi14020271_

Round 1
Reviewer 1 Report (New Reviewer)
This paper studied the piezoelectric energy harvesting device using galloping theoretically and experimentally. In particular, the theoretical considerations include the use of bivariate statistics to evaluate the dynamic response of energy harvesting devices. I would like to forward the following comments to the authors:
1. The title of this paper does not adequately represent the content of this paper. The author should name an appropriate title that reflects the content of this paper.
2. The overall content of this paper is unclear. The author should explain it more properly. In particular, do not leave the explanation to the references, but write it yourself.
3. In Figure 2, the force represents the force at what location and in what direction? And, considering that in energy harvesting of piezoelectric materials, a force(stress) is input and a voltage is output, isn't the variation in the results in Figure 2 too large?
4. In the explanation of lines 173 through 176, the author should explain what parameter determines this F value.
Author Response
Reviewer 1
This paper studied the piezoelectric energy harvesting device using galloping theoretically and experimentally. In particular, the theoretical considerations include the use of bivariate statistics to evaluate the dynamic response of energy harvesting devices. I would like to forward the following comments to the authors:
- The title of this paper does not adequately represent the content of this paper. The author should name an appropriate title that reflects the content of this paper.
Answer: Title changed.
- The overall content of this paper is unclear. The author should explain it more properly. In particular, do not leave the explanation to the references, but write it yourself.
Answer: Thank you for the comment. We have proofread the manuscript carefully and improved the writing.
- In Figure 2, the force represents the force at what location and in what direction? And, considering that in energy harvesting of piezoelectric materials, a force(stress) is input and a voltage is output, isn't the variation in the results in Figure 2 too large?
Answer: By the force in this study authors meant horizontal wind force acting on GPEH bluff body part, this force has oscillatory nature due to vortex shedding, see Figure 1. Now added in the text. What is “too large”?
- In the explanation of lines 173 through 176, the author should explain what parameter determines this F value.
Answer: F is a force. For the force description see answer above.
Reviewer 2 Report (New Reviewer)
Comments and Suggestions for Authors:
1. In Abstract, please add the novelties coming from your research. I really appreciate the fact that you refer it to the real application.
2. The good idea is to expand the bibliography with the part of applications of aero-flow piezoelectric energy harvesting systems discussing galloping and VIV phenomenon. I suggest following papers for citation, but it is not compulsory:
· Ceramic-based piezoelectric material for energy harvesting using hybrid excitation, Materials, 2021, 14(19), 5816
· Performance analysis of piezoelectric energy harvesting system, Advances in Science and Technology Research Journal, 2022, 16(6), pp. 179-185
3. For Figure 1 please provide its in higher resolution. After zooming, it is hardly visible to see the descriptions on it.
4. In the manuscript, I don’t see the detailed description of the experimental setup. Please provide it at least in table form.
5. Figure 2, the unit description for Force [N] is shifted (the same for Figure 6).
6. In Conclusions please provide the next steps of your research.
7. Please revise the text by the grammar and punctuation side.
After improving above described issues in the paper I’s like to sign my review report.
Author Response
Reviewer 2
- In Abstract, please add the novelties coming from your research. I really appreciate the fact that you refer it to the real application.
Answer: added.
- The good idea is to expand the bibliography with the part of applications of aero-flow piezoelectric energy harvesting systems discussing galloping and VIV phenomenon. I suggest following papers for citation, but it is not compulsory:
- Ceramic-based piezoelectric material for energy harvesting using hybrid excitation, Materials, 2021, 14(19), 5816
- Performance analysis of piezoelectric energy harvesting system, Advances in Science and Technology Research Journal, 2022, 16(6), pp. 179-185
Answer: added.
- For Figure 1 please provide its in higher resolution. After zooming, it is hardly visible to see the descriptions on it.
Answer: Figure 1 replaced
- In the manuscript, I don’t see the detailed description of the experimental setup. Please provide it at least in table form.
Answer: Experimental set up is illustrated in Fig 1, for other details reference given to Wang, J., Zhou, S., Zhang, Z., Yurchenko, D. (2019). "High-performance piezoelectric wind energy harvester with Y-shaped attachments." Energy Conversion and Management 181, pp. 645-652.
- Figure 2, the unit description for Force [N] is shifted (the same for Figure 6).
Answer: yes, it is.
- In Conclusions please provide the next steps of your research.
Answer: done.
- Please revise the text by the grammar and punctuation side.
Answer: Thank you for the comment. We have proofread the manuscript carefully and improved the writing.
Round 2
Reviewer 1 Report (New Reviewer)
1. The overall content of this paper is unclear. The author should explain it more properly. In particular, do not leave the explanation to the references, but write it yourself.
2. In Figure 2, considering that in energy harvesting of piezoelectric materials, a force(stress) is input and a voltage is output, isn't the variation in the results in Figure 2 too large?
3. In the explanation of lines 179 through 182, the author should explain what parameter determines this F value and how the value of this inflection point can be predicted.
4. Chapters 4 and 5 have the same name.
Author Response
- The overall content of this paper is unclear. The author should explain it more properly. In particular, do not leave the explanation to the references, but write it yourself.
I think that the author should clarify how this paper contributes to improving the properties of energy harvesting devices (e.g., increasing the power generation, suggesting less fragile conditions, etc.). Then, if possible, it should be discussed in terms of dimensionless parameters.
Reply: Thank you for the input. We have now included a paragraph in the introuction to highlight how this paper contributes to improving the design of energy harvesting devices.
New comments: Thanks for the correction. I think the sentence you added in the introduction is a good explanation of extreme value analysis. However, the author should explain in the conclusion how the results of "this study" will help in the design of energy harvesting devices concretely (For example, how to vary the extreme value distribution tail quantiles).
Answer: thanks for comment, rather than refer to quantile, we would rather refer to “characteristic” (or “design”) values, that have to be determined during design stage and therefore being an important input for a designer. Explanation added, see yellow mark-up.
- In Figure 2, considering that in energy harvesting of piezoelectric materials, a force(stress) is input and a voltage is output, isn't the variation in the results in Figure 2 too large?
First of all, I wrote "too large" because I am thinking in terms of the constitutive equation of piezoelectric materials (ε = sσ + dE, D = dσ + ϵE). When considering the constitutive equation of piezoelectric materials, force (stress) and voltage can be obtained in a one-to-one correspondence. Of course, since the entire device is deformed, I do not think that a simple one-to-one correspondence can be obtained, but even so, I felt that there was a large variation in Figure 2. In addition, I also thought that the relationship between force and voltage contains extra variation because it includes components other than excitation, i.e., components that are not related to destruction.
Reply: Thank you for the clarification. As you mentioned, the complex behaviour from the coupled interaction between wind, piezoelectric material and its supporting support could explain the relatively large variation in the results. It is important to note that the forces are still forces experienced by the piezoelectric material and therefore are essential to consider in its component design and the system design at large.
Thank you for your explanation. I am convinced by your explanation.
Answer: thanks a lot.
- In the explanation of lines 179 through 182, the author should explain what parameter determines this F value and how the value of this inflection point can be predicted.
Reply: It has been corrected. Please see preceding lines 166-171.
Surely, the author does add an explanation for the F value, but that is not what I really want to know. Related to my comment 1, if the F value can be controlled or predicted by using extreme value analysis, I think it will contribute to the design of appropriate energy harvesting devices. Therefore, I asked "what parameter determines this F value and how the value of this inflection point can be predicted".
Answer: well, this study has been focused on bivariate analysis, as a couple force-voltage has been chosen rather randomly, as again the focus was not on analyzing aerodynamics and VIV but on statistical methodology. Note that this study present experimental (not numerical or analytical) lab data, therefore it is not straightforward to analyze measured force and filter out VIV and other non-linear effects. As in any experiment there is a number of parameters, affecting measurements.
- Chapters 4 and 5 have the same name.
Reply: It has been corrected.
OK
Answer: thanks.
Reviewer 2 Report (New Reviewer)
Dear Authors,
All remarks have been introduced into the manuscript that I recommend it for its publishing in the present form. I wish you all the best for the upcoming 2023 year.
Reviewer
Author Response
thanks a lot, all comments addressed, best wishes in 2023!
Round 3
Reviewer 1 Report (New Reviewer)
Thank you for additional explanations. The author's answer was understandable and satisfactory.
This manuscript is a resubmission of an earlier submission. The following is a list of the peer review reports and author responses from that submission.
Round 1
Reviewer 1 Report
In this paper “Offshore galloping energy harvester’s experimental response statistics,” written by Oleg Gaidai, Yihan Xing, and Junlei Wang, galloping based energy harvester is studied based on a specific method called ACER2D. The reviewer cannot agree with the publication of this article in Micromachines. I have several concerns that should be addressed.
- Too many grammatical errors. English editing service is highly recommended.
- Also, there are a number of document errors. The authors should go over the manuscript very carefully to remove all those errors.
- The reviewer does not think that this article and its experimental work really deal with the “offshore” energy harvesters. The word “Offshore” should be excluded from the title.
- Abstract has to have just one paragraph. The numbered items shouldn’t be there. The first two sentences in abstract are not necessary. They should be in the introduction section. Lastly, abstract is not readable and it is hard to see what the authors are trying to point out.
- Overall, there is nothing new in the methodology (ACER2D), and the research outcomes are not scientifically significant and not sufficient enough for publication. Similar trends were already presented through the authors’ previous publications.
- Citing style is wrong. For example, in line 48, [2] has to be written as Mehmood et al. [2]. Also, there are too many self-citations. Particularly, more than 30 papers written by the first author were cited in this work, which is too much. Most of them are not directly related to the current work. They just used the ACER2D for analysis.
- Appendix is just a brief summary of ACER2D and there is no newly added thing in this part. Therefore, this part can be removed by citing one or two papers.
- In Figure 1, why is honey comb separated into several pieces? Also, providing a photo of the actual experimental apparatus will help readers have a better idea about the experiments.
- Figure 2, Figure 5, and Figure 6 are all force versus voltage graphs. Figure 2 is the results of experiments and the rest show the model predictions. Why is the trend of Figure 2 so different from Figures 5 and 6?
- Insert the reference for table 1. By the way, is Table 1 necessary? It is just used once in line 103, and as far as the reviewer see, it is not necessary for the paper.
- What are the meanings of the terms in equation (1)? Also, where do those values indicated in line 126-127 come from?
- What is the term “D” in line 143?
- Figure 3 is firstly mentioned in line 151, whereas Figure 2 in line 156, which is wrong.
- What does the y-axis represent in figure 4? What does ACER1D mean? What is the term k?
Author Response
Reviewer#1
In this paper “Offshore galloping energy harvester’s experimental response statistics,” written by Oleg Gaidai, Yihan Xing, and Junlei Wang, galloping based energy harvester is studied based on a specific method called ACER2D. The reviewer cannot agree with the publication of this article in Micromachines. I have several concerns that should be addressed.
- Too many grammatical errors. English editing service is highly recommended.
Answer: Thank you for the comment. We have now proof read carefully the manuscript and improved the grammar.
- Also, there are a number of document errors. The authors should go over the manuscript very carefully to remove all those errors.
Answer: Thank you for the comment. We have now proof read carefully the manuscript and improved the grammar.
- The reviewer does not think that this article and its experimental work really deal with the “offshore” energy harvesters. The word “Offshore” should be excluded from the title.
Answer: done.
- Abstract has to have just one paragraph. The numbered items shouldn’t be there. The first two sentences in abstract are not necessary. They should be in the introduction section. Lastly, abstract is not readable and it is hard to see what the authors are trying to point out.
Answer: The first two sentences and numbered items removed. The abstract is also now re-written to improved the quality.
- Overall, there is nothing new in the methodology (ACER2D), and the research outcomes are not scientifically significant and not sufficient enough for publication. Similar trends were already presented through the authors’ previous publications.
Answer: indeed, but the experimental data set analyzed in this study makes a difference.
- Citing style is wrong. For example, in line 48, [2] has to be written as Mehmood et al. [2]. Also, there are too many self-citations. Particularly, more than 30 papers written by the first author were cited in this work, which is too much. Most of them are not directly related to the current work. They just used the ACER2D for analysis.
Answer: why should reference be written as “Mehmood et al. [2].”? See e.g. https://www.mdpi.com/2072-666X/8/2/35 , https://www.mdpi.com/2072-666X/3/3/550 etc. for proper MDPI referencing style. Self-citations reduced.
- Appendix is just a brief summary of ACER2D and there is no newly added thing in this part. Therefore, this part can be removed by citing one or two papers.
Answer: done.
- In Figure 1, why is honey comb separated into several pieces? Also, providing a photo of the actual experimental apparatus will help readers have a better idea about the experiments.
Answer: there are 2 honey combs for stability, figure replaced now, so that included an actual photo.
- Figure 2, Figure 5, and Figure 6 are all force versus voltage graphs. Figure 2 is the results of experiments and the rest show the model predictions. Why is the trend of Figure 2 so different from Figures 5 and 6?
Answer: Fig 2 indeed indicates kind of bi-directional “trend”, however Figs 5,6 have nothing to do with that “trend”, as they display completely different thing. Fig 2 is phase space, Fig 5, 6 are probability contours – those are completely different issues.
- Insert the reference for table 1. By the way, is Table 1 necessary? It is just used once in line 103, and as far as the reviewer see, it is not necessary for the paper.
Answer: Table 1 removed.
- What are the meanings of the terms in equation (1)? Also, where do those values indicated in line 126-127 come from?
Answer: Eq. (1) is indicative about relationship between voltage and force. Re-written now.
- What is the term “D” in line 143?
Answer: term “D” definition added, see yellow mark-up.
- Figure 3 is firstly mentioned in line 151, whereas Figure 2 in line 156, which is wrong.
Answer: corrected.
- What does the y-axis represent in figure 4? What does ACER1D mean? What is the term k?
Answer: Decimal logarithmic scale on the vertical axis. Explanations added, see yellow mark-up.
Reviewer 2 Report
The authors propose a bivariate ACER2D technique that has the advantages of being able to study a wide range of coupled data and analyze shorter data. The method can be well used in various energy harvesting devices and has important engineering implications. There are some problems which need to be noticed
1. In line 45 of the manuscript, the authors claim that FIV is usually classified as VIV, galloping and wake galloping, however, wake galloping is only one type of wake induced vibration (WIV), which is a more complex type of FIV that may show characteristics of VIV and galloping as the flow velocity varies.
2. Abbreviations in the manuscript should be interpreted when they firstly appear, e.g. Energy Harvester (EH) should be interpreted on line 55 rather than on line 63.
3. The authors claim that the bluff body receiving the flow induced vibrations in Figure 1 is a circular-sectioned, but in fact, Figure 1 shows a rectangular body rather than a cylinder.
4. In line 105 of the manuscript, "If the wind speed it too high ......", perhaps the authors should have typed "is". Moreover, the author's description of Figure 4 involves "F=0.55N", however, the range of force in Figure 4 is only 0.02~0.09N. Careful proofreading and retouching of the manuscript is essential.
5. Please optimize Figure 4, it is so strange. Please confirm the completeness of Figure 6, it seems to me that it is missing a part of the legend.
Author Response
Reviewer#2
The authors propose a bivariate ACER2D technique that has the advantages of being able to study a wide range of coupled data and analyze shorter data. The method can be well used in various energy harvesting devices and has important engineering implications. There are some problems which need to be noticed
- In line 45 of the manuscript, the authors claim that FIV is usually classified as VIV, galloping and wake galloping, however, wake galloping is only one type of wake induced vibration (WIV), which is a more complex type of FIV that may show characteristics of VIV and galloping as the flow velocity varies.
Answer: corrected.
- Abbreviations in the manuscript should be interpreted when they firstly appear, e.g. Energy Harvester (EH) should be interpreted on line 55 rather than on line 63.
Answer: corrected.
- The authors claim that the bluff body receiving the flow induced vibrations in Figure 1 is a circular-sectioned, but in fact, Figure 1 shows a rectangular body rather than a cylinder.
Answer: Fig 1 replaced with proper one.
- In line 105 of the manuscript, "If the wind speed it too high ......", perhaps the authors should have typed "is". Moreover, the author's description of Figure 4 involves "F=0.55N", however, the range of force in Figure 4 is only 0.02~0.09N. Careful proofreading and retouching of the manuscript is essential.
Answer: corrected.
- Please optimize Figure 4, it is so strange. Please confirm the completeness of Figure 6, it seems to me that it is missing a part of the legend.
Answer: Fig 4 title updated to include more detail. Text, explaining Fig 4 added, see yellow mark-up. Fig. 6 confirmed correct.
Reviewer 3 Report
See attachement.

Author Response
Review
“Offshore galloping energy harvester’s experimental response statistics” by O. Gaidai et al. shows a statistical method using bivariate ACER2D method.
The structure of the paper is sound however the information is confusing presented and there are claims that are not clearly supported by data. All these make the paper very difficult to follow, to understand what their contribution and novelty is. The contribution needs major revision for clarity.
Comments and questions:
Abstract
C1. “…. have been a lot of new advancements in harnessing electrical energy via piezoelectric materials.” Add to introduction references and/or info to support this statement.
Answer: added
C2. “ Energy harvesters constitute an important part of modern offshore green energy engineering “ Add to introduction references and/or info to support this statement.
Answer: added
C3. “ Advantage of the proposed methodology is that relatively short experimental data record can still yield meaningful statistical and design results, “ It is not clear where in the contribution this statement is supported, for ex. how short the data record should be, how the results are influence by this.
Answer: explanation added, see yellow mark-up.
C4. It is stated that “ energy harvesting devices may be damaged by strong wind forces due to the wake galloping. Safety and reliability are important engineering concerns “ This statement of ‘strong wind’ and safety & reliability’ are not coming through as results and/or analysis in the contribution. So, where is the supported input to these?
Answer: sentence corrected.
C5. There are three key points mentioned but which is the supporting information in the contribution to connect the bivariate method with strong wind forces & Safety and reliability?
Answer: “strong wind forces” term now removed, as generally extreme EH response being in focus.
Introduction
C6. Above comments.
C7. “ The macro-scale wind energy “ What is this? I assume you refer to harvester scale and not wind energy scale or? If yes, then it should be ‘…above cm3 scale harvesters…’ as ‘macro-scale’ harvesters are wind turbines.
Answer: “macro-scale” removed.
C8. For statements in lines 36-40 and lines 42-43 it should be good to add some references.
Answer:
C9. “ Unfortunately, mechanical vibration is usually irregular and not continuous.” Wrong formulation: mechanical vibration can be regular (e.g. industrial pumps, motor). Rephrase.
Answer:Sentence removed.
C10. “ but few of them paid attention to some practical design issues like EH extreme response and fatigue life. Studying extreme performance of the EH and avoiding damaging working conditions are important parts ”Does this mean that you are studying these? If yes, where are these presented and analyzed? If not, then why mention?
Answer: yes this paper studies EH extreme response.
C11. “ There was nearly no research focusing on experimental testing of dynamic performance of a wind-induced vibration energy harvesting with respect to its reliability. ” Does this mean that you are studying reliability of harvester for dynamic performance? If yes, where are this presented and analyzed? If not, why mention?
Answer: yes, this paper studies extreme EH response, which is a key part of reliability study.
C12: the same question as C11 for safety & reliability in this statement “ but few studies were focused on the EH extreme values analysis, and thus safety and reliability of its working conditions. ”
Answer: yes, it is exactly the topic of this study.
C13. “ This paper studies extreme value statistics of a GPEH system dynamic response ” For what purpose? What is new/ different from other similar studies? Add this information.
Answer: added, see yellow mark-up.
Methodology and analysis
C14. “ can produce stable incoming flow ” I assume you want to say ‘uniform’ flow.
Answer: yes, corrected.
C15. Is this “ wind speed was within a range of 1 ≤ U ≤ 6 m/s ” giving EH extreme response?“
Answer: yes, for example 6 m/s may induce VIV strong enough to cause EH damage, added now.
C16. What is “ clamped capacitance……..”?
Answer: see e.g. https://www.ingentaconnect.com/contentone/dav/aaua/1971/00000025/00000001/art00014
Results
C17. What is “ Stationary voltage ”? Is open circuit voltage? If yes, what is the internal impedance of the oscilloscope?
Answer: replaced with “time series”.
C18. Line 170: “ and non-linear statistical approach is required ” Do you use it?
Answer: Sentence removed.
C19. “ laboratory measured data set is quite limited ” What is minimum number of data that you can still get trustful analysis results?
Answer: now stated in abstract, see yellow mark-up, its few orders of magnitude less than target probability.
Discussion
C20. “ introduction of general reliability methodology ” Is reliability for your statistical methodology or for harvester design? I have difficulties to see this analysis for any of these. Comment and make clear what is your message with this article.
Answer: Sentence removed.
Conclusions
C21. “ This paper studied energy harvester dynamic response ” Open-circuit voltage is not the only important parameter, you need to measure also the short-circuit current of the harvester (If any of these has low values, you cannot start harvester electronics thus no useful harvester), voltage after electronics.
Answer: this study is focused on bivariate statistical methodology, thus choosing any other alternative couple of responses will be possible. Now added, see yellow mark-up.
C21. “ during operation in random wind conditions. ” Where is the analysis for ‘random’ wind conditions analysis? As “ six different wind speeds ” is not random.
Answer: random comes through wind speed PDF in Fig 3, as constant wind speeds then being plugged into this distribution. Now added, see yellow mark-up.

Reviewer 4 Report
The article is devoted to the study of the reliability of galloping based piezoelectric energy harvester based on the analysis of experimental data by the statistical method ACER2D (two dimensional averaged conditional exceedance rate). In general, the article is well written, but there are two points.
1. The article considers a device with dimensions of 200 × 25 × 0.5 mm^3. The scope of the journal says that its subject is devoted to the study of micro/nano devices. The scale of the device shown in the article clearly does not fall into this range. Might be worth considering another journal. And with a bias in reliability and statistics.
2. The article is not very large, but 92 sources of literature were used in it. Moreover, in the text you can find such extensive references as “[18]-[42]”. It seems to me that this is redundant. Either it is worth describing each source in more detail or reducing their number. In addition, it seemed to me that some literary sources were not used at all in the text.
A number of typos were also noticed, as well as inconsistency in the introduction of designations. For example, the designation GPEH first appeared on line 52, and its meaning was only revealed on line 74.
Author Response
Reviewer#4
The article is devoted to the study of the reliability of galloping based piezoelectric energy harvester based on the analysis of experimental data by the statistical method ACER2D (two dimensional averaged conditional exceedance rate). In general, the article is well written, but there are two points.
- The article considers a device with dimensions of 200 × 25 × 0.5 mm^3. The scope of the journal says that its subject is devoted to the study of micro/nano devices. The scale of the device shown in the article clearly does not fall into this range. Might be worth considering another journal. And with a bias in reliability and statistics.
Answer: as mentioned, this study may be seen in perspective to micro/nano-wind energy harvesters, as advocated reliability approach is general and applicable to any scale. Now text added, see yellow mark-up.
- The article is not very large, but 92 sources of literature were used in it. Moreover, in the text you can find such extensive references as “[18]-[42]”. It seems to me that this is redundant. Either it is worth describing each source in more detail or reducing their number. In addition, it seemed to me that some literary sources were not used at all in the text.
Answer: Thank you for the comment. We have now reviewed the references and removed some citations.
A number of typos were also noticed, as well as inconsistency in the introduction of designations. For example, the designation GPEH first appeared on line 52, and its meaning was only revealed on line 74.
Answer: corrected.
Round 2
Reviewer 1 Report
The authors have not addressed the reviewer's comments adequately. Furthermore, the manuscript does not deserve a publication due to lack of methodological merit and scientific merit as well as a poor quality of presentation. There are still too many meaningless self-citation, for example, see "[17]-[31]" in line 115. Overall, the reviewer cannot agree with its publication in materials.
Reviewer 3 Report
Thanks to the authors for their answer but unfortunately there are two main issues:
- You still must explain what is new/ special with your methodology in comparison to literature.
- You cannot just remove sentences/ words instead of answering the question and give details in the text.
More remarks are in the attached Word document.

Reviewer 4 Report
Work has improved markedly.
In general, we can agree that the technique described in the article is applicable to devices of various scales. To make your position more weighty, you can cite works or reviews dedicated to micro/nano wind energy harvesters.
However, a serious flaw remained in the work in the form of an excessive number of self-citations. In the previous version, with 92 references, the percentage of self-citations was about 40%. Now, when there are 41 sources, self-citations have increased to 56%. It might be worth looking for a balance.